Insights and progress on the biosynthesis, metabolism, and physiological functions of gamma-aminobutyric acid (GABA): a review

Zhang Qingli 1
Zhu Lei 1
Li Hailong 1
Chen Qu 1
Li Nan 2
Li Jiansheng 3
Zhao Zichu 1
Xiao Di 1
Tang Tingting 1
Bi Chunhua 1
Zhang Yan 1
Zhang Haili 1
Zhang Guizhen 1
Li Mingyang 1
Zhu Yanli 4
Zhang Jingjing 1 oojingjingoo@126.com
Kong Jingjing 1 18642919681@163.com
1 Department of Medicine, Qingdao Binhai University , Qingdao, Shandong , China
2 Department of Rehabilitation, Qingdao Binhai College Affiliated Hospital , Qingdao , China
3 Department of Nephrology, Gansu Provincial Hospital of Traditional Chinese Medicine , Lanzhou, Gansu , China
4 College of Life Sciences and Engineering, Northwest Minzu University , Lanzhou , China
Uversky Vladimir
Electronic publication date: 2024 Dec 16
Publication date: 2024
Volume: 12
Electronic Location ID: e18712
Received 2024 Aug 19; Accepted 2024 Nov 24
Copyright: © 2024 Zhang et al.
Copyright year: 2024
Copyright holder: Zhang et al.
License: This is an open access article distributed under the terms of the Creative Commons Attribution License, which permits unrestricted use, distribution, reproduction and adaptation in any medium and for any purpose provided that it is properly attributed. For attribution, the original author(s), title, publication source (PeerJ) and either DOI or URL of the article must be cited.
License URL: https://creativecommons.org/licenses/by/4.0/

Keywords: γ-Aminobutyric acid, Neurotransmitter, Biosynthesis, Metabolism, Physiological function

Funding: Shandong Provincial Natural Science Foundation ZR2022QD153, ZR2023MH098 and ZR2024QH643 National Natural Science Foundation of China 82360884 This work was supported by the Shandong Provincial Natural Science Foundation (Nos. ZR2022QD153, ZR2023MH098, ZR2024QH643) and the National Natural Science Foundation of China (No. 82360884). The funders had no role in study design, data collection and analysis, decision to publish, or preparation of the manuscript.

==============================
GABA (γ-aminobutyric acid) is a non-protein amino acid that occurs naturally in the human brain, animals, plants and microorganisms. It is primarily produced by the irreversible action of glutamic acid decarboxylase (GAD) on the α-decarboxylation of L-glutamic acid. As a major neurotransmitter in the brain, GABA plays a crucial role in behavior, cognition, and the body’s stress response. GABA is mainly synthesized through the GABA shunt and the polyamine degradation pathways. It works through three receptors (GABAA, GABAB, and GABAC), each exhibiting different pharmacological and physiological characteristics. GABA has a variety of physiological roles and applications. In plants, it regulates growth, development and stress responses. In mammals, it influences physiological functions such as nervous system regulation, blood pressure equilibrium, liver and kidneys enhancement, hormone secretion regulation, immunity enhancement, cancer prevention, as well as anti-aging effects. As a biologically active ingredient, GABA possesses unique physiological effects and medicinal value, leading to its widespread application and substantially increased market demand in the food and pharmaceutical industries. GABA is primarily produced through chemical synthesis, plant enrichment and microbial fermentation. In this review, we first make an overview of GABA, focusing on its synthesis, metabolism, GABA receptors and physiological functions. Next, we describe the industrial production methods of GABA. Finally, we discuss the development of ligands for the GABA receptor binding site, the prospects of GABA production and application, as well as its clinical trials in potential drugs or compounds targeting GABA for the treatment of epilepsy. The purpose of this review is to attract researchers from various fields to focus on GABA research, promote multidisciplinary communications and collaborations, break down disciplinary barriers, stimulate innovative research ideas and methods, and advance the development and application of GABA in medicine, agriculture, food and other fields.

Introduction

Aminobutyric acid is a general term for amino acids containing amino and butyric acid groups. The main isomers are α-aminobutyric acid (AABA, also known as homoalanine), β-aminobutyric acid (BABA) and γ-aminobutyric acid (GABA, also called 4-aminobutanoic acid, piperidic acid, and piperidinic acid). Studies have shown that AABA levels are associated with depressive symptoms, tuberculosis, Reye’s syndromes, and many other diseases (Adachi et al., 2019; Oketch-Rabah et al., 2021; Wang et al., 2020b). Additionally, AABA has been studied as a general biomarker for sepsis, multi-organ failure, alcoholic liver injury and malnutrition (Wang et al., 2020b). However, BABA has not been found to be correlated with human diseases, and it is a natural product of plants controlled by their immune systems. GABA is a four-carbon non-proteinogenic water-soluble amino acid (C4H9NO2) that is widely present in prokaryotes and eukaryotes in the form of free amino acid (Wang et al., 2020a; Yuan et al., 2019) (Fig. 1).

Figure 1 Chemical formulas of three isomers of aminobutyric acid (AABA, BABA, GABA).

AABA, α-aminobutyric acid; BABA, β-aminobutyric acid; GABA, γ-aminobutyric acid.

The rapid accumulation of GABA in plants is a response to abiotic stresses (such as hypoxia, heat, cold, drought, and mechanical wounding) or biotic stresses (such as wounding due to herbivory and infection) (Bown & Shelp, 2016; Wang et al., 2020b). GABA was regarded as a plant and microbial metabolite and was synthesized for the first time in 1883 (Oketch-Rabah et al., 2021). Natural GABA was first discovered in potatoes by Steward in 1949, followed by its identification in mammalian brains, and subsequently in animals as well as several other organisms, including bacteria and fungi (Steward, Thompson & Dent, 1949; Fromm, 2020; Roberts & Frankel, 1950). GABA is also a neurotransmitter found across a range of invertebrate phyla (e.g., arthropods, echinoderms, annelids, nematodes, and flatworms) (Miller, 2019). This review mainly focuses on GABA from mammals and plants. As an inhibitory neurotransmitter, GABA is most widely distributed in the central nervous system in mammals, but it is also expressed in small amounts in many peripheral tissues (e.g., liver, kidney, stomach, small intestine, and reproductive organs). GABA serves as the primary neurotransmitter in at least one-third of the neurons in the central nervous system of rats, and it accounts for nearly 30% of human brain neurons (Oketch-Rabah et al., 2021). The distribution of GABA varies across different regions of the rat brain, with the substantia nigra exhibiting the highest concentration, followed by the globus pallidus, nucleus accumbens, deep cerebellar nuclei, caudate nucleus, and cerebellar cortex (Tepper & Lee, 2007).

GABA is considered one of the inhibitory neurotransmitters involved in a variety of metabolic activities, such as depression, anxiety and stress management, hypertension, diabetes, cancer activity, oxidative, inflammatory, microbial, and allergic resistance, and protection of the liver, kidney, and intestine. As a result, it is widely used in the functional food, pharmaceutical, and agricultural products industries (Dahiya, Manuel & Nigam, 2021; Ngo & Vo, 2019; Wei et al., 2022).

In this review, we first provide a brief introduction to the three isomers of aminobutyric acid (AABA, BABA and GABA) and their functions, focusing on the discovery, development and importance of GABA. We also summarize the biosynthesis and metabolism of GABA, as well as the three GABA receptors (GABAA, GABAB and GABAC) and their physiological functions in detail. Additionally, we briefly outline the role of GABA in regulating plant growth, development and stress, while placing particular emphasis on its effects on mammalian physiological functions. Subsequently, we review three commonly used methods for GABA enrichment preparation, highlighting their advantages and disadvantages. Finally, we analyze the potential of microbial fermentation for GABA enrichment and discuss the development and application of ligands for the GABA receptor binding site. New clinical trials targeting GABA for the treatment of epilepsy are briefly discussed, along with a detailed examination of potentially effective drugs or compounds targeting GABA for epilepsy treatment.

The purpose of this review is, firstly, to summarize the research progress and organize the related knowledge, allowing readers to quickly and comprehensively understand the research results concerning GABA in terms of its synthesis, metabolism, mechanism of action, receptor types, physiological functions and biosynthesis. Secondly, this review aims to attract researchers from different fields to focus on GABA research, promote communication and cooperation among neuroscience, pharmacology, botany, microbiology and other multidisciplinary disciplines, break down the boundaries between disciplines, and stimulate new research ideas and methods.

This review provides a detailed description of the three isomers of aminobutyric acid (AABA, BABA, and GABA), as well as a systematic summary of the history of GABA’s discovery and development, important milestones, and its physicochemical properties. Currently, there is a significant gap in the searchable literature concerning this section.

The main limitations of this review are as follows: First, due to the wide range of GABA-related research fields and limited space, this review only provided a brief overview of the role of GABA in regulating plant growth, development, and stress, lacking a more comprehensive and in-depth analysis. Second, some earlier references are difficult to access in their original language; however, it is worth noting that many scholars generally recognize the findings of these studies. Additionally, epilepsy, as one of the significant brain disorders, has been selected for discussion in new clinical trials targeting GABA across various conditions. On this basis, this review also discusses in depth potential drugs or compounds that target GABA, using epilepsy as an example. This will help enhance our understanding of the importance of GABA in the treatment of a wide range of diseases.

Survey/search methodology

The authors conducted a comprehensive and systematic search across PubMed, Web of Science, GenMedical, and Google Scholar. The research utilized key terms such as “Gamma-Aminobutyric Acid”, “GABA”, “neurotransmitter”, “biosynthesis”, “metabolism”, “physiological functions”, and “epilepsy”, both as subject terms and free words. The initial screening of the literature focused on titles, which was followed by a secondary screening on abstracts and keywords. In addition, ClinicalTrials.gov, Clinicaltrialsregister.eu, and the China Clinical Trials Registry were searched for ongoing studies targeting GABA for epilepsy. Ultimately, the full texts of relevant articles were obtained for further evaluation. Additionally, the authors retrieved references from pertinent studies to identify other eligible research. The target audiences for this research are diverse, comprising academics, physicians, pharmaceutical technology developers, health nutritionists, pharmaceutical producers, and innovators.

Timeline of development and key milestones to gaba

Discovery and isolation: In the 1880s, scientists succeeded in synthesizing GABA, which was thought to be a metabolite of plants and microorganisms (Oketch-Rabah et al., 2021). Natural GABA was first discovered in potatoes by Steward in 1949 (Steward, Thompson & Dent, 1949). Discoveries in the animal brain: In 1950, GABA was discovered to be present in mammalian neural tissues by Roberts & Frankel (1950), and Awapara (1950), respectively. Although the physiological activity of GABA was not yet understood, Roberts and Frankel demonstrated that GABA in mouse brains is synthesized through alpha-decarboxylation, using glutamate (Glu) as a substrate in the presence of glutamate decarboxylase (GAD) (Roberts & Frankel, 1950; Heli et al., 2022). This marked an important beginning in the study of GABA and laid the foundation for subsequent research on its role in the nervous system (Fig. 2).

Figure 2 Timeline of GABA-related development and key milestones.

Exploration as a neurotransmitter: In 1959, Takahashi et al. (1959) studied the effects of GABA on the cerebral cortex. In 1961, Curtis, Phillis & Watkins (1961) found that GABA had an inhibitory effect on spinal cord neurons, providing further evidence for GABA’s function as a neurotransmitter. In 1963, Krnjevic suggested that GABA might be a central neurotransmitter, however, this idea was not widely recognized at first (Krnjevic & Phillis, 1963). In 1967, Krnjevic and Schwartz demonstrated that GABA mimics endogenous neurotransmitters in regulating neurons in the mammalian cerebral cortex, hyperpolarizes the membrane potential, and causes a decrease in membrane resistance (Krnjevic & Schwartz, 1967).

Discovery and characterization of GABA receptors: From the 1970s to 1980s, scientists discovered and characterized GABAA and GABAB receptors, identifying them as ionotropic and metabotropic G protein-coupled receptors, respectively (Curtis & Johnston, 1974; Sytinsky, 1978; Bowery et al., 1980; Olsen, 1981). The molecular cloning of GABAA receptor subunits further revealed the diversity and structure of these receptors (Schofield et al., 1987).

Follow-up researches and application expansion: After the 1980s, advances in structural biology techniques helped elucidate the detailed structure of GABA receptors and revealed the association of GABA dysfunction with various psychiatric and neurological disorders, providing important information for drug design (Mehta & Ticku, 1999; Olsen & Sieghart, 2009; Sigel & Steinmann, 2012; Miller & Aricescu, 2014; Pin & Bettler, 2016).

Since GABA was recognized as an important neurotransmitter in the 1940s, its research has progressed through multiple stages, ranging from basic biological discovery to clinical drug development. Through in-depth studies of its structural biology, scientists have identified the link between GABAergic dysfunction and numerous diseases, driving the development of GABAergic drug therapies as well as the creation and application of functional foods. In recent years, the use of new technologies has further expanded the research horizons of the GABA system, providing more possibilities for future therapeutic strategies.

Physicochemical properties of gaba

GABA has a IUPAC (International Union of Pure and Applied Chemistry) name of 4-aminobutanoic acid, a CAS (Chemical Abstracts Service) number of 56-12-2, and a UNII (Unique Ingredient Identifier) code number of 2ACZ6IPC6I (Oketch-Rabah et al., 2021). Its molecular formula is C4H9NO2, its relative molecular weight is 103.1. The amino group of GABA is located on its γ-carbon. GABA is a white to light yellow crystalline substance that can crystallize as white leaflets (in methanol-ether) or needles (in water-ethanol). It has a slightly bitter taste and exists as a zwitterion at physiological pH, possessing both positive and negative charges, with pKa values of 10.56 and 4.03, respectively (Oketch-Rabah et al., 2021; Shelp, Bown & McLean, 1999; Xu, Wei & Liu, 2017). However, GABA is not spinable. It is highly soluble in water (130 g/100 ml at 25 °C), minimally soluble in ethanol and acetone, and insoluble in cold ethanol, benzene, and ether. Its melting point is 202 °C, and it can decompose above this temperature to form pyrrolidone and water (Zhichao, 2016).

As a derivative of an amino acid, GABA not only possesses the common properties of amino acid but also exhibits unique characteristics that enable it to undergo specific chemical reactions (e.g., hydrocarbonization and acylation with chloride reactions). GABA can react with ninhydrin to form a purple compound, which has a maximum absorption at 570 nm. Additionally, GABA can react with dansyl chloride (DNS-Cl) under alkaline conditions to produce a fluorescent and stable DNS-GABA, which absorbs light in the UV region. These two reactions can be utilized for the detection of GABA (Roberts & Frankel, 1950).

Gaba biosynthesis and metabolic pathways

GABA is typically synthesized by glutamate decarboxylase (GAD; EC4.1.1.15), which utilizes pyridoxal 5′-phosphate (PLP) as a cofactor. This enzyme catalyzes the irreversible α-decarboxylation of L-glutamate to GABA, releasing CO2 during the process (Fig. 3).

Figure 3 GABA synthesis: decarboxylation of L-Glutamate by glutamate decarboxylase (GAD) and pyridoxal-5-phosphate (PLP).

GABA shunt

The GABA shunt pathway was first described in 1970 in a study involving guinea pig cells (Balazs et al., 1970). It is named as “GABA shunt” because it bypasses two steps of the tricarboxylic acid (TCA) cycle and is associated with electron transport in the respiratory chain (Fig. 4) (Anju, Moothedath & Rema Shree, 2014). The GABA shunt is a conserved pathway in both prokaryotes and eukaryotes, and it is closely linked to the antioxidant system (Anju, Moothedath & Rema Shree, 2014). As an energy source, GABA is considered an essential defense strategy, providing NADH and/or succinate to the mitochondrial electron transport chain in the presence of impaired respiration and TCA cycle activity, as well as increased reactive oxygen species (Carillo, 2018; Carillo et al., 2019). The GABA shunt pathway is not equivalent to the conversion of a-ketoglutamate to succinate in the TCA cycle. In the TCA cycle, the metabolic conversion produces one ATP and one NADH, whereas the GABA shunt produces only one NADH. Since the GABA shunt produces less energy and only 8–10% of TCA cycle activity occurs through this shunt, its primary function is thought to be the synthesis of GABA (Patel, Balazs & Richter, 1970; Watanabe et al., 2002).

Figure 4 Interconnection of the GABA shunt and polyamine degradation (PA) pathway in metabolic regulation.

Orange arrows indicate the GABA shunt; Green arrows indicate the PA pathway. GABA shunt: GABA biosynthesis begins with the transamination of α-ketoglutarate (α-KG), catalyzed by glutamate dehydrogenase (GDH), to produce glutamate. Subsequently, the irreversible decarboxylation at the α- carbon of glutamate, catalyzed by glutamic acid decarboxylase (GAD), yields GABA. This reaction consumes a proton and releases carbon dioxide and can occur in both the mitochondria and the cytoplasm. GABA catabolism is then catalyzed by GABA transaminase (GABA-T), converting GABA into succinic semialdehyde (SSA). SSA is subsequently oxidized by succinate-semialdehyde dehydrogenase (SSADH) and enters the tricarboxylic acid (TCA) cycle as succinate. PA pathway: In this pathway, glutamate is converted to α-KG by GABA-T. This α-KG then undergoes a series of enzymatic reactions to produce glutathione and, ultimately, ornithine. Ornithine is transported to the cytoplasm, where it is converted into putrescine by ornithine decarboxylase (ODC). Through further enzymatic reactions, putrescine is converted to spermidine and spermine. Putrescine, spermidine, and spermine are then converted by diamine oxidase (DAO) and polyamine oxidase (PAO), respectively, to produce 4-aminobutyraldehyde. This 4-aminobutyraldehyde is dehydrogenated by 4-aminobutyraldehyde dehydrogenase (AMADH) to form GABA. The PA pathway intersects with the GABA shunt, and their metabolites eventually enter the TCA cycle.

Three enzymes are involved in the GABA shunt: cytoplasmic glutamate decarboxylase (GAD), mitochondrial GABA transaminase (GABA-T), and succinate semialdehyde dehydrogenase (SSADH), respectively (Li et al., 2020; Sarasa et al., 2020). The first step in GABA anabolism involves glutamate dehydrogenase catalyzing the transamination reaction of α-ketoglutarate to produce glutamate. The second step is the irreversible decarboxylation of the alpha carbon of glutamate to GABA, catalyzed by GAD, which consumes a proton and releases CO2. GABA catabolism begins with GABA-T catalyzing the conversion of GABA into succinic semialdehyde (SSA). SSA is then oxidized by SSADH and enters the TCA cycle as succinate (Sarasa et al., 2020; Shelp, Bown & McLean, 1999). In plants, the GABA shunt is considered necessary for maintaining the TCA cycle under both stress and non-stress conditions (Bown & Shelp, 2020). While GABA shunt enzymes are primarily located in mitochondria, their subcellular localizations vary across different organisms (GAD is located in the cytoplasmic, GABA-T and SSADH are mainly found in mitochondria, but in yeast, SSADH is located in the cytoplasm) (Cao et al., 2013; Clark et al., 2009b).

GAD

GAD is widely present in microorganisms, animals and plants, and both its species and structure exhibit variations across species (Cui et al., 2020). In mammals, there are two different isoforms of GAD: GAD67 and GAD65, which have molecular weights of 67 and 65 kDa, respectively. These isoforms are encoded by GAD1 and GAD2 genes. In humans, both genes are located on chromosomes 2 and 10 (Erlander et al., 1991; Kaufman, Houser & Tobin, 1991; Soghomonian & Martin, 1998; Watanabe et al., 2002). Although GAD67 and GAD65 catalyze the same decarboxylation reaction, they fulfill different functional roles. GAD67 synthesizes GABA in the brain, while GAD65 is the predominant isoform found in pancreatic cells. GAD67 is more prevalent during early development and is present in the terminals and cell bodies of mature neurons, potentially contributing to the intracellular pool of GABA in nonsynaptic cells. In contrast, GAD65 is usually expressed later in development and is mainly restricted to nerve terminals (Pierobon, 2021; Watanabe et al., 2002). Furthermore, the two isoforms of GAD differ among species. Human β-cells express only GAD65, while mice predominantly express GAD67 and rats express both GAD65 and GAD67 (Kanaani et al., 2015).

GABA-T and SSADH

GABA-T belongs to the pyridoxal phosphate-dependent transaminase III family and plays a key role in the catabolism of GABA by catalyzing the formation of SSA and alanine from pyruvate and GABA (Heli et al., 2022). Both GABA-T and SSADH are localized in mitochondria; GABA-T catalyzes reversible reactions, whereas SSADH typically exhibits greater activity than GABA-T with a low affinity for SSA. Therefore, GABA-T and SSADH work together to facilitate GABA catabolism (Watanabe et al., 2002). Studies have indicated that the optimal pH for GAD in plant tissues is around 5.8, while the optimal pH for GABA-T is between 9.0 and 9.5 (Clark et al., 2009a; Liao et al., 2017; Sun et al., 2022). The enzymatic activity of SSADH is significantly influenced by the presence of Cu2+ at pH 7.5 and 37 °C (Heli et al., 2022).

Polyamine degradation pathway

Polyamines are aliphatic amines containing two or more amines (-NH2) or imines (-NH-) groups. They are widely distributed in both prokaryotic and eukaryotic cells. Common polyamines include putrescine, spermidine and spermine (Cha et al., 2014). The polyamine degradation pathway (PA pathway) involves the production of γ-aminobutyraldehyde which is catalyzed by diamine oxidase (DAO) and polyamine oxidase (PAO). This intermediate is then dehydrogenated by γ-aminobutyraldehyde dehydrogenase (AMADH) to form GABA, which ultimately intersects with the GABA branch and participates in TCA cycle metabolism (Shelp, Mullen & Waller, 2012). Putrescine serves as the central substance in the PA pathway, and DAO and PAO are key enzymes. The polyamine degradation pathway acts as a supplementary route for GABA production. However, the level of putrescine-derived GABA in the adult rat brain is negligible, and only about 2% to 3% of GABA in the adult mouse brain may come from putrescine (Konishi, Nakajima & Sano, 1977; Seiler, Bink & Grove, 1980; Watanabe et al., 2002). Approximately 30% of the GABA accumulation in germinating broad beans under non-stress and anoxic stress conditions is attributed to polyamine degradation (Yin, Cheng & Fang, 2018). It is speculated that the response of putrescine to GABA involves both gene-dependent and non-dependent process (Shelp et al., 2012).

Gaba receptors

In mammals, GABA exerts its neurotransmitter effects by binding to specific receptors, which are classified as GABAA, GABAB, and GABAC. These receptors differ in their sensitivities to agonists and antagonists (Bormann, 2000; Heli et al., 2022). GABAA is an ionotropic receptor that can be activated by high concentrations of GABA, functioning as a fast-acting ligand-gated chloride channel that evokes synaptic inhibition (Adler, 2014). GABAB receptors, on the other hand, are slow-acting metabolic G protein-coupled receptors that primarily regulate synaptic transmission and participate in various brain functions, such as recognition, learning, memory, and anxiety (Printsev, Curiel & Carraway, 2017; Wei et al., 2022). GABAC is also an ionotropic receptor with a morphology similar to that of GABAA receptors. In adult brains, GABA mainly acts on GABAA and GABAB receptors, thereby influencing the rhythmic activity generated in neural networks (Wei et al., 2022; Cuypers, Maes & Swinnen, 2018).

GABAA receptors

The GABAA receptor family represents a class of ligand-gated ion channels that are permeable to Cl− and HCO3−, and have been extensively studied due to their significant physiological and therapeutic roles. GABAA receptors consist of a heterodimeric transmembrane protein complex made up of three of the seven subunit families (α1–6, β1–3, γ1–3, δ, ε, θ, and π) (Mehta & Ticku, 1999). There is approximately 60–80% amino acid homology within each subunit class and about 30% homology between different classes (Watanabe et al., 2002). A central ion-conducting pore is formed by five adjacent subunits. Each receptor subunit has one extracellular structural domain (ECD), one transmembrane structural domain (TMD), and one intracellular structural domain (ICD) (Fig. 5A). The ECDs consist of β-sheets and contribute to agonist binding sites. The TMDs consist of pore-forming α-helices, while the structurally variable ICDs are involved in receptor assembly, transport, and aggregation.

Figure 5 Schematic representation of the GABAA receptor and its subunits.

(A) The generic topological structure of a GABAA receptor subunit. The mature GABAA receptor comprises 450 amino acid residues. It possesses a large hydrophilic extracellular domain (ECD) at the N-terminus, four hydrophobic transmembrane domains (TMDs) (including TM1-TM4) in the middle, and a smaller intracellular domain (ICD) at the C-terminus. TM1 is connected to TM2 by a short intracellular loop; TM2 is connected to TM3 by a short extracellular loop; and TM3 is connected to TM4 by a long intracellular loop, which is phosphorylatable. (B) GABAA receptors are heteropentameric chloride-ion channels. They consist of five subunits, typically arranged in a heterodimeric fashion, forming a cylindrical channel. At the center of this channel is a pore permeable to chloride ions; the GABAA receptor exerts an inhibitory effect in the central nervous system by modulating the cell membrane’s depolarization. GABAA receptors are formed from 16 subunits: α1–6, β1–3, γ1–3, δ, ε, θ, and π. Functional GABA receptors contain at least one α-subunit, one β-subunit, and one γ-subunit. The most common pentameric combinations are 2α2β1γ, 2α1β2γ, and 1α2β2γ. 2α2β1γ is most frequently expressed in the central nervous system. (C) Distribution of GABA and benzodiazepine (BZD) binding sites in the GABAA receptor. Two GABA binding sites are located at the β+/α− interface, while one BZD binding site is located at the α+/γ− interface. Additionally, more than 10 potential ligand-binding sites are distributed throughout the receptor.

Since the GABAA receptor is a pentamer composed of various subunit assemblies, it should theoretically be possible to form a large number of receptor isoforms. However, the diversity of receptors is highly constrained, and mechanisms controlling the differential assembly and subcellular targeting of receptor subtypes remain unclear (Rudolph & Knoflach, 2011). Studies indicate that a functional GABA receptors contains at least one α, one β, and one γ, with the most common pentameric combinations being 2α2β1γ, 2α1β2γ, and 1α2β2γ. Among these, 2α2β1γ is the most frequently expressed in the central nervous system (Watanabe et al., 2002) (Fig. 5B).

A mature GABAA receptor subunit consists of approximately 450 amino acid residues. Its N-terminal features a large hydrophilic extracellular structural domain (ECD), while the middle section contains four hydrophobic transmembrane structural domains (TMDs: TM1–TM4). The C-terminal contains a small extracellular domain. TM1 and TM2 are connected by a short intracellular loop, TM2 and TM3 are linked by a short extracellular loop, and TM3 and TM4 are joined by a long intracellular loop that can be phosphorylated (Ghit et al., 2021) (Fig. 5A).

The GABA binding site is located at the β+/α- junction (Watanabe et al., 2002) (Fig. 5C). In addition to the GABA binding sites, there are over 10 potential ligand binding sites that are presumably distributed at various locations on the GABAA receptor. These sites are important drug targets and contribute to the receptor a highly complex pharmacological profile (Chua & Chebib, 2017). The GABAA receptor family serves as the site of action for many clinically important drugs, including benzodiazepines (BZD), barbiturates, anesthetics, neurosteroids, and convulsants. These drugs can affect GABAA receptor function by inducing or stabilizing conformational changes in the receptor, as well as binding to different metabolic or non-metabolic sites to produce physiological or clinical effects, ultimately influencing GABAA receptor function (Mirheydari et al., 2014; Crocetti & Guerrini, 2020). The development of ligands for GABAA receptor binding sites aims to the advancement of new therapeutic drugs.

GABAB receptors

GABAB is a receptor that reduces the release of norepinephrine and is insensitive to bicuculline and isoglutamate. It was first described in the 1980s by Bowery et al. (1980). Nakayasu et al. (1993) reported the purification of putative GABAB receptor proteins for the first time. The GABAB-Rla and GABAB-Rlb proteins both belong to the G protein-coupled receptor family. The two are N-terminal splice variants containing 960 and 844 amino acids, respectively, and the cDNA sequences encoding both have been isolated and reported (Kaupmann et al., 1997).

The functional GABAB receptor is known to be a specialized heterodimer formed by the co-assembly of GABAB1 and GABAB2 subunits, and each subunit is considered as incapable of signaling alone (Shaye et al., 2021; Xu et al., 2014). Each subunit consists of a large extracellular structural domain (Venus flytrap, VFT), a seven-transmembrane structural domain, and an intracellular C-terminus (Fritzius et al., 2022; Terunuma, 2018). Each VFT contains LB1 and LB2 structural domains or lobes, where LB1 is positioned above LB2, and the VFT can exists in either an open or closed form Frangaj & Fan (2018) (Fig. 6). The VFT is a common structural feature of all class C G protein-coupled receptors, and many selective splice variants exist for GABAB1 (e.g., R1a, R1b, R1c, R1e, R1j, R1k, R1l, R1m, and R1n). In the human CNS, the main subtypes are R1a and R1b, which are distinguished by a pair of sushi structural domains that exist in R1a but absent in R1b. This likely relates to the presence of the sushi structural domain, which mediates multiple protein interactions of GABAB1 (Benke et al., 1999; Terunuma, 2018). Although GABAB2 shares 54% similarity with GABAB1, only the VFT structural domain of GABAB1 can bind to ligands, while the VFT of GABAB2 cannot bind to any known ligands (Xu et al., 2014). However, the heptahelical structural domain of the GABAB2 subunit contains binding sites for metastable regulators that can affect the affinity of ligands for binding to the GABAB1 subunit. The interaction between the GABAB1 and GABAB2 subunits occurs through the coiled helical structural domain at their C-terminus (Terunuma, 2018).

Figure 6 Heterodimeric and tetrameric structures of the GABAB receptor.

GABAB receptors exist in equilibrium as heterodimers, tetramers (dimers of dimers), and octamers (tetramers of dimers). Heterodimers are stabilized by strong noncovalent interactions, while higher-order oligomers are formed through weaker, and possibly transient, dimeric interactions. GABAB receptor tetramers are formed through GABAB1-GABAB1 interactions between dimers. Each subunit comprises a large extracellular Venus Flytrap (VFT) domain, a seven-transmembrane domain, and an intracellular C-terminus. The GABAB1 and GABAB2 subunits interact via their C-terminal coiled-coil domains; neither subunit can signal independently. The VFT domain at the N-terminus of GABAB1 is responsible for ligand binding (such as GABA and baclofen), whereas the VFT domain of GABAB2 does not bind any known ligand. Positive allosteric modulators can bind to thetransmembrane domain of GABAB2 to potentiate the agonist’s effects.

GABAB dimers, tetramers or higher order oligomers have been detected in heterologous systems. GABAB receptors exist in an equilibrium of three aggregates (Xu et al., 2014). GABAB receptors mediate slow and persistent inhibition by activating Galphai/o-type G proteins, leading to the liberation of Gbetagamma subunits that activate inwardly rectifying K+ channels, inactivate voltage-gated Ca2+ channels, or inhibit adenylate cyclase (Craig & McBain, 2014; Terunuma, 2018). The GABA analogs baclofen and 3-aminopropylphosponous acid (3APPA; CGP27492) are typical agonists of GABAB, whereas saclofen, phaclofen and 2-hydroxysaclofen act as antagonists of GABAB receptors (Jembrek & Vlainic, 2015).

GABAB receptors not only mediate the actions of major inhibitory neurotransmitters but also associate with neurological and gastrointestinal disorders, as well as mental health issues. GABAB has been identified as a multi-drug therapeutic target for conditions such as muscle spasms, pain, alcohol addiction, schizophrenia, and gastroesophageal reflux disease. As a muscle relaxant, Lioresal® (baclofen) is the only FDA-approved drug that selectively targets GABAB for clinical application (Shaye et al., 2020, 2021).

GABAC receptors

Johnston and his colleagues discovered that partially folded GABA analogue cis-4-aminocrotonic acid selectively activates a third class of GABA receptors in the mammalian central nervous system. Tentatively named GABAC in 1984, these receptors are chlorine pores (Bormann, 2000). The GABAC receptor is insensitive to both bicuculline (a characteristic ligand of GABAA receptors) and baclofen (a characteristic ligand of GABAB receptors) (Jembrek & Vlainic, 2015). GABAC is highly expressed in many parts of the brain, including the superior colliculus, cerebellum, hippocampus, and predominantly in the retina (Gavande et al., 2011; Lukasiewicz, 1996). The structure of the GABAC receptor is similar to that of the GABAA receptor, which is a pentameric channel consisting of ρ (ρ1–3) subunits. However, there are significant differences in the biochemistry, pharmacology and physiology between these two receptors (Chebib & Johnston, 1999). For example, the GABAC receptor is more sensitive than the GABAA receptor; it has smaller currents and does not desensitize (Watanabe et al., 2002). It should be noted that the amino acid sequence of the ρ subunit of the GABAC receptor is only 30–38% similar to that of the GABAA receptor subunit (Bormann, 2000). In addition, they differ in gene distribution: the ρ1–2 and ρ3 genes are located on human chromosomes 6 and 3, respectively, while the α, β, γ, and ε genes are distributed on chromosomes 4, 5, 15, or X (Bormann, 2000; Watanabe et al., 2002). The GABAC ρ1 subunit not only plays an important role in inhibitory pathways and sensory processing in the retina and spinal cord but also acts as an inhibitory modulator in olfactory bulb neurons, affecting olfactory signaling processes (Chen et al., 2007). A study has shown that motor function after stroke can be significantly improved by inhibiting GABAC receptors. Thus targeting the ρ subunit may provide a new delayed therapeutic option for post-stroke recovery (Van Nieuwenhuijzen et al., 2021).

Physiological effects and applications of gaba

Regulation of plant growth, development, and stress by GABA

Plants can spontaneously produce and secrete GABA, which is present in varying concentrations in plant embryos, seeds, roots, stems, leaves, flowers, fruits, and other organs. The concentration of GABA may vary significantly depending on the external environment (Heli et al., 2022; Ramesh et al., 2017). For example, the insect bites on Arabidopsis thaliana can evoke rapid local and systemic responses to plant cell cycle signals, leading to an increase in GABA levels in the plants. At the same time, excessive intake of GABA (as a neurotrophic substance) by insects can also affect their growth and development (Kiep et al., 2015). In addition, GABA rapidly accumulates in plants undergoing stresses such as hypoxia, mechanical stress, exposure to phytohormones, water stress, cold stress and heat stress. This accumulation plays an important role in reducing reactive oxygen species levels, redistributing carbon and nitrogen resources, regulating pH, promoting plant growth and development, and increasing tolerance to oxidative stress (Yuan et al., 2017; Guo et al., 2023; Heli et al., 2022; Yuan et al., 2023). Furthermore, GABA and GABA receptor agonists and antagonists also affect plant growth. Nowadays, GABA is widely used as an exogenous additive to cope with various (e.g., salt and drought) stresses and to promote growth and development, and thereby improving plant yield and quality (Heli et al., 2022).

Effects of GABA on physiological functions in mammals

GABA, as a major inhibitory neurotransmitter, has effects that extend beyond suppressing anxiety, relieving depression, and providing sedation and sleep aid. It also has various physiological functions, including lowering blood pressure, strengthening liver and kidney functions, regulating hormone secretion, enhancing immune resistance to cancer, reducing asthma symptoms, controlling obesity, providing anti-aging benefits, improving diabetes management, and enhancing learning and memory abilities.

Regulating the nervous system

As a major inhibitory neurotransmitter, GABA can effectively alleviate or directly suppress anxiety by blocking the reception of anxiety-related information in the central nervous system and binding to anxiety-reducing receptors in the brain (Jiang et al., 2020). Furthermore, the excitation-to-inhibition (E/I) ratio plays a crucial role in normal brain development and function, and imbalances in this ratio can lead to neurodevelopmental abnormalities (Culotta & Penzes, 2020). In mature neurons, GABA primarily serves an inhibitory role. However, in the developing nervous system, the high intracellular concentrations of chloride ions enable GABA to have a depolarizing effect, which promotes the entry of calcium ions into the cell, activating a series of signaling cascades that ultimately promote neuronal growth and synaptic development (Ben-Ari et al., 2012; Markicevic et al., 2020). In the glial cells of the grasshopper, both phasic and tonic GABAergic signaling are present. This signaling plays a unique role in regulating olfactory adaptation and neuronal senescence (Cheng et al., 2024). Additionally, GABA supplementation has been shown to be capable of improving depression (Yin et al., 2018). When a neuron is excited, the binding of GABA to GABAA receptors increases the permeability of chloride ions, leading to a negative change in transmembrane potential and decreased neuronal excitability, thereby providing a sedative effect (Jacob, Moss & Jurd, 2008). Sleep disorders are closely related to decreased levels of serotonin (5-HT) and GABA in the central nervous system, and GABA supplements have been shown to be helpful for both humans and animals to fall asleep more easily (Byun et al., 2018; Jeong et al., 2021; Sun et al., 2020; Wang et al., 2019; Yamatsu et al., 2015). Therefore, GABA plays an important role in regulating the nervous system by inhibiting anxiety, alleviating depression, calming the body and aiding sleep.

Reducing blood pressure

GABA can regulate blood pressure by modulating both the central and peripheral sympathetic nervous systems (Abe et al., 1995). GABA is associated with a transient and moderate decrease (less than a 10% change) in blood pressure in hypertensive animals and humans (Oketch-Rabah et al., 2021; Shimada et al., 2009; Takahashi, Sumi & Koshino, 1961). Specifically, when the signal of raised blood pressure is transmitted to the central nervous system, GABA inhibits the action of angiotensin-converting enzyme by blocking the formation of angiotensin II, thereby effectively lowering blood pressure (Koh et al., 2023). Given GABA’s blood pressure-lowering effect, related functional foods for reducing blood pressure are currently being extensively researched and developed. GABA-rich natural foods, such as tea, brown rice, and other items, as well as GABA-producing microorganisms (e.g., Chlorella, Lactobacillus, Bacillus subtilis) and fermented dairy products and legumes, have been shown to be capable of preventing and controlling the development of hypertension (Abe et al., 1995; Bao & Chi, 2016; Lim et al., 1990; Oda et al., 2014; Shimada et al., 2009; Suwanmanon & Hsieh, 2014). Therefore, the utilization of GABA and its source foods in the field of blood pressure reduction appears promising.

Strengthening the liver and kidney

GABA has significant protective effects on the liver. It not only reduces hepatocyte apoptosis in acute liver injury by attenuating inflammation and oxidative stress but also alleviates ethanol- and CCl4-induced liver injury by maintaining intracellular polyamine levels (Chen et al., 2022; Hata et al., 2019; Norikura et al., 2007). In the kidneys, GABA’s protective effects are also significant; it prevents renal vasoconstriction and ischemia, as well as toxin-induced injury and glycerol-induced acute renal failure (Dahabiyeh et al., 2020; Nasiri et al., 2017). Furthermore, in a model of major nephrectomy, GABA treatment has been shown to protect the kidneys from tubular fibrosis and atrophy, as well as ischemia-reperfusion injury (Prud’homme, Kurt & Wang, 2022). Thus, GABA shows promising potential in preventing renal failure.

Regulating hormone secretion

GABA has regulatory effects on several hormones in the body, including growth hormone, thyroid hormone, and glucagon. Specifically, GABA can regulate growth hormone levels in humans, rats, monkeys, and piglets via the hypothalamus or the pituitary gland (End et al., 2005; Oketch-Rabah et al., 2021; Powers, 2012). Regarding the regulation of thyroid hormones, experiments have shown that the expression of key genes for thyroid hormone synthesis (e.g., TG, TPO, and NIS) is significantly elevated in hypothyroid mice consuming GABA, leading to increased blood levels of T4 and T3 (Yang et al., 2019). Additionally, GABA synthesized by pancreatic β-cells acts as a paracrine and autocrine signal that inhibits glucagon secretion by α-cells and regulates pancreatic islet homeostasis (Kim et al., 2019; Menegaz et al., 2019). In summary, GABA plays an important role in the regulation of the synthesis and secretion of several hormones.

Enhancing immunity and preventing cancer

Levels of GABAA receptors or other signaling components are often upregulated in cancer cells, which implies that manipulating GABAA receptor activity may reduce tumor growth. For example, the GABAA receptor allosteric agonist nebuta has been shown as capable of inhibiting the growth and metastasis of experimental colon cancer (Young & Bordey, 2009). Additionally, GABA secreted by B cells can bind to and activate GABAA receptors on nearby immune cells, thereby inhibiting the infiltration and activity of cytotoxic T cells and macrophages, thereby attenuating the anti-tumor response (Zhang et al., 2021; Tian & Kaufman, 2023). Concurrently, GABA also reduces cAMP levels through GABAB receptors and prevents the activation of PKA, thus blocking the migration of tumor cells (Ortega, 2003). Some studies have reported that GABAB receptor agonists and GABA inhibit cell proliferation and migration in gastric, colon, and malignant hepatocytes (Kanbara et al., 2018). Systemic administration of the gamma-aminobutyric acid transporter-1 inhibitor NO-711 is prophylactic and therapeutic for paclitaxel-induced thermal nociceptive hypersensitivity and cold allodynia in some patients with dose-limiting painful peripheral neuropathy caused by paclitaxel during cancer treatment (Masocha & Parvathy, 2016). Furthermore, GABA administration has been found as functional in increasing human IgA levels, and thereby enhancing immunity under stress conditions (Abdou et al., 2006; Sahab et al., 2020). In animal studies, GABA supplementation has been shown as useful in promoting jejunal secretory IgA secretion, enhancing intestinal mucosal immunity in piglets (Zhao et al., 2020). Although GABA shows potential inhibitory effects on cancer, most of these results are derived from in vitro cellular and animal experiments, and its actual effects on humans still need to be confirmed by further clinical studies.

Anti-aging

GABA levels decline with age, and the decline is a trigger for cognitive decline and brain aging in humans. GABA is helpful in the long-term development and understanding of neuronal aging (Florey, 1991). Studies have shown that GABA can improve the fertility of adult silkworms and exhibits anti-aging potential by regulating energy balance, lowering carbohydrate and lipid levels, improving antioxidant capacity, and regulating the expression of mRNA associated with longevity-related genes (Tu et al., 2022). Neuronal proteomic studies in young and aging animals revealed that the channel protein TMC-1 achieves its anti-aging function by regulating GABA signaling, and the TMC-1-GABA-PKC signaling axis is neuroprotective under aging and disease conditions (Wu et al., 2022). A therapeutic serum containing GABA and a variety of peptides with neurotransmitter inhibitory and cell signaling properties has been shown to be capable of improving the appearance of expression lines under the eyes and on the face, including crow’s feet and forehead lines (Draelos, Kononov & Fox, 2016). In conclusion, GABA shows great potential in anti-aging effects for both skin and the nervous system, and therefore can regulate lifespan.

Other functions

GABA also has the ability to reduce asthma (Park et al., 2013; Nayak & An, 2022), control obesity (Xia et al., 2021), improve diabetes (He et al., 2016) and enhance learning and memory (Liu et al., 2015).

Enrichment preparation of GABA

The levels of GABA found in natural foods are far from satisfying the physiological needs of the human body (Jiang et al., 2023). Traditionally, GABA is obtained from natural plants, but the supply is insufficient to meet market demand. Currently, the major methods for large-scale industrial production of GABA include chemical synthesis, plant enrichment, and microbial fermentation (Heli et al., 2022; Luo et al., 2021).

Chemical synthesis

GABA is produced via chemical synthesis using pyrrolidone, butyrolactone, thionyl chloride, butyric acid, and ammonia as raw materials to open the ring through chemical reactions (Heli et al., 2022). The advantage of the chemical synthesis method is its relatively high yield, while the disadvantages include difficulty in controlling the operational process, which can be dangerous, and high equipment costs. Furthermore, the chemical synthesis of GABA generates chemical residues and causes environmental pollution. Therefore, the industrial production of GABA is limited.

Plant enrichment

Plant enrichment of GABA primarily utilizes the GABA shunt pathway (the main pathway) and the PA pathway (Liu et al., 2022). Plants that undergo hypoxic stress, low-temperature stress, high salt osmotic stress, and high-temperature thermal stress will experience effects that compel them to adapt to drastic changes in their external environments. In response, they spontaneously regulate enzyme activity within their systems, and an increase in enzyme activity may lead to a significant accumulation of GABA through both pathways (Cheng et al., 2018; Shi et al., 2010). In recent years, non-thermal techniques such as high hydrostatic pressure, ultrasound, and electrolyzed water have been demonstrated as safe and effective inducers of GABA enrichment in plants (Heli et al., 2022). Compared to the chemical synthesis method, the plant enrichment method is simpler and safer to operate, although it has the disadvantages of low production efficiency, difficulties in isolation, and low GABA content obtained, indicating that this method still needs improvement.

Microbial fermentation

Lactic acid bacteria (LAB), Aspergillus, Monascu, Haematococcus, yeasts, and Bacillus can produce GABA (Luo et al., 2021; Yogeswara, Maneerat & Haltrich, 2020). For instance, Lb. brevis NCL912, screened from Chinese kimchi, produced GABA titers of 103.72 g/L (Wang et al., 2018). E. avium G-15, isolated from carrot leaves, produced 115.7 g/L of GABA in a tank fermenter (Tamura et al., 2010). Sodium ions (Na+) can inhibit the growth and metabolism of microorganisms, reducing the quality of fermentation products. In the sodium-ion-free fermentation, L. brevis CD0817 achieved GABA production levels of up to 331 ± 8.3 g/L (Li et al., 2023). Aspergillus oryzae NSK, screened from Malaysian soy sauce koji, produced 354.08 mg/L of GABA using cane molasses as the fermentation substrate (Hajar-Azhari et al., 2018). The maximum yield of GABA from Monascus sanguineus, isolated from spoiled pomegranate when coconut oil cake was used as the substrate was predicted to be 15.53 mg/gds (Dikshit & Tallapragada, 2015). The yield of GABA from M. purpureus CCRC 31615 increased to 1,267.6 mg/kg with the addition of sodium nitrate. The maximum GABA yield in microalga Haematococcus pluvialis was 38.57 mg (Ding et al., 2019). The highest GABA yield from yeast Kluyveromyces marxianus C21 in okara was found to be 4.31 g/L (Zhang et al., 2022). Additionally, the yield of GABA from Bacillus cereus KBC under optimized conditions was 3,393.02 mg/L (Wan-Mohtar et al., 2020). Moreover, engineered strains such as Escherichia coli and Corynebacterium glutamicum can also produce GABA (Luo et al., 2021).

Discussions

GABA is a non-protein amino acid that occurs naturally in microorganisms, plants, and animals (Ngo & Vo, 2019). GABA is mainly synthesized through the GABA shunt and the PA pathway (Heli et al., 2022). The physiological functions of GABA differ between the animal and plant kingdoms. In plants, GABA regulates growth, development and stress responses (e.g., mechanical stress, stimulation, hypoxia, darkness, and drought). As the primary inhibitory neurotransmitter of the central nervous system, GABA is present in approximately 25–50% of mammalian neurons (Miri et al., 2023). However, the functions of GABA extend beyond the treatment of various psychiatric and neurological disorders (e.g., epilepsy treatment, anxiety suppression, depression relief, sedation and sleep assistance) to include lowering blood pressure, enhancing liver and kidney function, regulating hormone secretion, boosting immune resistance and fighting cancer, reducing asthma, controlling obesity, anti-aging and anti-skin aging, improving diabetes, and enhancing learning and memory abilities. Given GABA’s beneficial effects on human health, it has been used as a bioactive ingredient in food and clinical applications. GABA has been employed as a dietary supplement in the United States, China, Japan, and most of Europe (Tian et al., 2022). Additionally, GABA, its derivatives, and its receptors are widely utilized in clinical studies for the treatment of epilepsy, insomnia, hypertension, and stress, as well as an ergogenic substance to increase growth hormone levels (Oketch-Rabah et al., 2021; Sun & Wang, 2023; Durand et al., 2023).

GABA production methods are mainly chemical synthesis, plant enrichment, and microbial fermentation (Heli et al., 2022). GABA prepared by chemical synthesis is known to have poor safety, as it can easily be contaminated, making it unsuitable for usage in food or medicine. In contrast, GABA produced by living organisms (both plants and microorganisms) can meet the health demands of organisms with almost no side effects. Consequently, research on GABA-rich plants and microorganisms has become a hot topic, with many researchers focusing on the development of GABA-rich functional foods. Among these methods, microbial fermentation is preferred for commercial use because it is more effective, less costly, and more environmentally friendly. Currently, although both bacteria and fungi serve as good sources for GABA production, the genus Lactobacillus has been the most widely used for this purpose (Oketch-Rabah et al., 2021).

GABA interacts with three types of receptors: GABAA, GABAB and GABAC (Bormann, 2000). GABAA and GABAC are fast-acting ligand-gated chloride channel receptors, while GABAB is a slow-acting metabolic G protein-coupled receptor (Kim et al., 2018). Each receptor possesses different binding sites, thus leading to distinct pharmacological and physiological characteristics. The development of ligands for the GABA receptor binding site has contributed to a better understanding of various physiological functions and pathological processes, as well as to the development of new therapeutic drugs.

Epilepsy is a prevalent chronic brain disease characterized by recurrent seizures (Chen et al., 2023). These seizures are caused by abnormal discharges of neurons in the brain, which are transient and sudden, that can occur at any age and hence affect individuals throughout their lifespan (Chen et al., 2024). GABA plays a crucial role in the treatment of epilepsy and can effectively control seizures by regulating neuronal activity. A clinical trial assessing the effects of GABA on epilepsy and ADHD in children is currently in Phase IV (NCT04144439). Gabapentin and pregabalin, which are structurally similar to GABA, do not act directly on GABA receptors. Instead, these two drugs are used primarily for the treatment of partial-onset epilepsy by binding to the α2δ subunit of voltage-gated calcium channels, which reduces the release of excitatory neurotransmitters and indirectly enhances GABAergic inhibition. Both drugs have completed phase III clinical trials (NCT00603473; NCT00448916). Pregabalin has been approved by the U.S. Food and Drug Administration (FDA) as an adjunctive treatment for partial-onset seizures in adults (Cross, Viswanath & Sherman, 2024). Sodium valproate has a molecular structure similar to that of the neurotransmitter GABA, and the addition of lamotrigine to valproate is currently being investigated to evaluate its effects on epilepsy in children, with clinical trials in phase IV (NCT05881928). ETX101 consists of a non-replicating recombinant adeno-associated virus serotype 9 vector designed for the delivery of a GABAergic regulatory element and an engineered transcription factor capable of enhancing the transcription of the SCN1A gene. ETX101 has shown promising results in clinical epilepsy studies and is currently undergoing clinical trials (Lersch et al., 2023; NCT06112275). Muscimol, which resembles a natural brain chemical called GABA, has been shown as capable of reducing seizures in rats. A clinical trial is currently in phase I to assess the safety and efficacy of injecting muscimol into the brain to control refractory epilepsy (i.e., seizures that are frequent and persist despite treatment) (NCT00005925).

There are many potential possible drugs or compounds that could target GABA for the treatment of epilepsy. Recent research has focused on epilepsy associated with disorders such as Tuberous Sclerosis Complex (TSC), Lennox-Gastaut syndrome, focal refractory epilepsy, and Dravet syndrome. GABAA receptors play a key role in inhibitory neurotransmission; therefore, drugs targeting GABAA receptors have received extensive research attentions in the treatment of epilepsy. GABAA receptor modulators can be categorized into benzodiazepines and non-benzodiazepines.

Representative drugs of the benzodiazepine class include clobazam, clonazepam, and diazepam. These are three drugs share similarities in the mechanisms for treating epilepsy: they can bind to benzodiazepine receptors, promoting the binding of GABA to GABA receptors. This process increases the frequency of chloride channel openings, leading to hyperpolarization of the neuronal membrane and inhibition of neuronal firing. By enhancing the inhibitory effects of GABA, these drugs can reduce neuronal excitability and exert an antiepileptic effect. Clobazam is an oral 1,5-benzodiazepine that enhances GABAA receptor function and reduces the frequency and severity of seizure, completed a Phase IV clinical trial in 2018 (NCT02726919). It is used globally for the treatment of many types of epilepsy and is approved in the United States for the treatment of Lennox-Gastaut syndrome (Gauthier & Mattson, 2015). Clonazepam orally disintegrating tablets (Klonopin®) have been used in the acute treatment of epileptic seizures and are indicated for both seizures and panic attacks. More than 50% of seizures stop within 10 min, and it is as effective as diazepam rectal gel (Penovich et al., 2024; Troester, Hastriter & Ng, 2010; Cloyd et al., 2021). Diazepam rectal gel was approved by the U.S. Food and Drug Administration in 1997 for patients with acute epilepsy aged 2 years and older. However, rectal administration has limitations. Diazepam nasal spray was also approved in 2020 for the treatment of patients with acute epilepsy aged 6 years and older (Penovich et al., 2024).

Non-benzodiazepines, such as phenobarbital, ganaxolone, and darigabat, also target GABAA receptors. Phenobarbital effectively prolongs the opening time of the chloride channel by acting on the GABAA receptor, significantly enhancing the inhibitory effect of GABA. It can effectively prevent subsequent seizures through intravenous administration and has been used to treat neonatal epilepsy, having already completed phase III clinical trials (NCT04320940). Ganaxolone is a neuroactive steroid that positively allosterically modulates GABAA receptors, effectively reducing seizures, with a mechanism of action similar to that of phenobarbital. Currently, ganaxolone is in phase II clinical trials, showing potential therapeutic value for TSC (NCT04285346). Darigabat, a positive allosteric modulator of GABAA receptors, has demonstrated significant efficacy in a mesial temporal lobe epilepsy (a preclinical model of medically refractory focal epilepsy). A Phase II clinical trial (NCT04244175) is currently underway to evaluate the efficacy and safety of daligabat in patients with medically refractory focal seizures (Gurrell et al., 2022).

In addition to research on drugs targeting the GABAA receptor for the prevention and treatment of epilepsy, research is actively underway on a wide range of drugs or compounds targeting the GABAA, GABAB and GABAC receptors for the prevention and treatment of a variety of disorders, such as Parkinson’s disease, muscle spasms, depression and others. These studies are aimed at developing more effective drugs with fewer side effects, with a view to providing a wider range of therapeutic options in clinical applications.

Since the interaction between GABA receptors and ligands can effectively combat a variety of diseases and facilitate pharmacological research and drug development. It has become a major focus of research. However, the mechanism by which GABA controls the differential assembly of receptor subtypes and subcellular targeting remains unclear. The number, location, and roles of GABA binding sites have not yet been systematically and thoroughly investigated. The study of the multiple physiological functions of GABA continue to be the primary focus of current research. Additionally, the development of GABA-enriched functional foods has emerged as a hot research topic. This research primarily involves the development of high GABA-producing strains, improving the stability of GABA during storage, and developing new enrichment techniques without compromising the quality of food or other active ingredients. Currently, the microbial fermentation method for GABA production holds broad prospects, with bacteria, fungi and engineered bacteria being widely studied. An increasing number of studies on GABA will provide more solutions for drug development and disease treatment.

Conclusions

GABA is a naturally occurring non-protein amino acid found in microorganisms, plants and animals, synthesized through various pathways. It exhibits different physiological functions in both the animal and plant kingdoms and provides numerous benefits for humans. The primary methods for GABA production include chemical synthesis, plant enrichment, and microbial fermentation. Among these, microbial fermentation has become the preferred choice for commercial production due to its high efficiency, low cost, and reduced environmental impact. In terms of physiological mechanisms, GABA interacts with three types of receptors: GABAA, GABAB and GABAC. The development of ligands targeting the binding sites of these receptors not only contributes to in-depth studies of related diseases but also facilitates the research and development of new drugs. In the field of epilepsy treatment, GABA plays a critical role, and various drugs are available that control seizures by targeting the GABA system through different mechanisms. Clinical trials related to these drugs are at various stages, indicating promising prospects for development. While the interaction between GABA receptors and ligands has become a hot research topic, there is still a need for in-depth exploration of how GABA controls subtype assembly, subcellular targeting mechanisms, and the specific functions of binding sites. Additionally, the development of GABA-rich functional foods is emerging as a key area of research, with the microbial fermentation method showing great promise in this field. In conclusion, GABA is expected to have many positive impacts on health, daily life and industrial development across various fields, including medicine, food, agriculture, and cosmetics.

Abbreviations

GABA Gamma-Aminobutyric Acid

GAD glutamic acid decarboxylase

AABA α-aminobutyric acid

BABA β-aminobutyric acid

IUPAC International Union of Pure and Applied Chemistry

CAS Chemical Abstracts Service

UNII Unique Ingredient Identifier

DNS-Cl dansyl chloride

PLP pyridoxal 5′-phosphate

TCA tricarboxylic acid

SSADH succinate semialdehyde dehydrogenase

GABA-T GABA transaminase

PA pathway polyamine degradation pathway

DAO diamine oxidase

PAO polyamine oxidase

AMADH γ-aminobutyraldehyde dehydrogenase

ECD extracellular structural domain

TMD transmembrane structural domain

ICD intracellular structural domain

BZD benzodiazepines

LAB Lactic acid bacteria

TSC Tuberous Sclerosis Complex

SSA succinic semialdehyde

α-KG α-ketoglutarate

ODC ornithine decarboxylase

GDH glutamate dehydrogenase

Additional Information and Declarations

Competing Interests

Author Contributions

Data Availability

The authors declare that they have no competing interests.

Qingli Zhang conceived and designed the experiments, performed the experiments, analyzed the data, prepared figures and/or tables, and approved the final draft.

Lei Zhu conceived and designed the experiments, performed the experiments, analyzed the data, prepared figures and/or tables, and approved the final draft.

Hailong Li conceived and designed the experiments, performed the experiments, prepared figures and/or tables, and approved the final draft.

Qu Chen performed the experiments, prepared figures and/or tables, authored or reviewed drafts of the article, and approved the final draft.

Nan Li performed the experiments, authored or reviewed drafts of the article, and approved the final draft.

Jiansheng Li performed the experiments, authored or reviewed drafts of the article, and approved the final draft.

Zichu Zhao performed the experiments, authored or reviewed drafts of the article, and approved the final draft.

Di Xiao performed the experiments, authored or reviewed drafts of the article, and approved the final draft.

Tingting Tang performed the experiments, authored or reviewed drafts of the article, and approved the final draft.

Chunhua Bi performed the experiments, authored or reviewed drafts of the article, and approved the final draft.

Yan Zhang analyzed the data, authored or reviewed drafts of the article, and approved the final draft.

Haili Zhang analyzed the data, authored or reviewed drafts of the article, and approved the final draft.

Guizhen Zhang analyzed the data, authored or reviewed drafts of the article, and approved the final draft.

Mingyang Li analyzed the data, authored or reviewed drafts of the article, and approved the final draft.

Yanli Zhu analyzed the data, authored or reviewed drafts of the article, and approved the final draft.

Jingjing Zhang conceived and designed the experiments, prepared figures and/or tables, and approved the final draft.

Jingjing Kong conceived and designed the experiments, prepared figures and/or tables, and approved the final draft.

The following information was supplied regarding data availability:

This is a literature review.

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
