# Peer review of "Insights and progress on the biosynthesis, metabolism, and physiological functions of gamma-aminobutyric acid (GABA): a review"

_PeerJ, doi:10.7717/peerj.18712_

## Round 0.1 · original submission · Major Revisions

Please address concerns of both reviewers and amend your manuscript accordingly.

Reviewer 1 ·

Basic reporting

The manuscript entitled “Insights and progress on the biosynthesis, metabolism, and physiological functions of GammaAminobutyric Acid (GABA)” in which the authors gave an overview of GABA, focusing on its synthesis, metabolism, GABA receptors, and physiological functions. Also, they discussed the development of ligands for GABA receptor binding site, the prospects of GABA production and application.
The work is understandable and important. However, this paper suffers from some shortcomings that if modified would make the manuscript suitable for publication in PeerJ, Journal.
Shortcomings:
1- In Title page, please add * to distinguish the corresponding authors.
2- Please add list of abbreviation.
3- In introduction section,
• Please the aim of study in end of introduction part.
• Please add in detail the need of reviewing the GABA to fill the gap of knowledge in this part.
4- Please discuss in brief, new clinical trial of targeting GABA in different diseases especially brain disorders.
5- Please discuss in detail, potential possible drugs or compounds that could target GABA and used in treatment of different diseases especially brain disorders.
6- Please add a summarized conclusion in the end of conclusions and future perspectives.
7- Please add the limitation of this study.

Experimental design

OK

Validity of the findings

4- Please discuss in brief, new clinical trial of targeting GABA in different diseases especially brain disorders.
5- Please discuss in detail, potential possible drugs or compounds that could target GABA and used in treatment of different diseases especially brain disorders.
6- Please add a summarized conclusion in the end of conclusions and future perspectives.
7- Please add the limitation of this study.

Reviewer 2 ·

Basic reporting

In this review paper, the authors effectively summarize and discuss recent insights and advancements in the biosynthesis, metabolism, and physiological functions of Gamma-Aminobutyric Acid (GABA). As a crucial neurotransmitter in the central nervous system, GABA primarily serves inhibitory functions, reducing neuronal excitability and calming neural activity to maintain balance in the brain. The authors begin with a comprehensive overview of GABA, addressing its synthesis, metabolism, receptor interactions, and physiological roles. They also explore industrial production methods for GABA and discuss the development of ligands targeting GABA receptor binding sites, highlighting the potential applications of GABA in enhancing human health and commercial production. Overall, the manuscript exhibits clear logic and offers a detailed review of GABA. However, in its current state, it is not suitable for publication in this journal and will require revisions before resubmission.

Experimental design

GABA has been implicated in various cellular processes related to cancer, such as cell proliferation, migration, and invasion. The effects of GABA in cancer biology are still being explored and may vary depending on the cancer type. Summarizing the relationship between GABA and cancer could significantly enrich the review and provide valuable insights for potential therapeutic applications targeting GABAergic signaling.

Validity of the findings

1. A timeline outlining the development and key milestones related to GABA would enhance the reader's understanding of its historical context and relevance.

2. The following references may provide valuable background information and should be considered for inclusion during revision:
A: Anxiolytics for Bronchodilation: Refinements to GABAA Agonists for Asthma Relief
B: GABAA receptor subtype modulators in medicinal chemistry: an updated patent review (2014-present)
C: Aging and GABA, PMID: 29905530
D: Phasic/tonic glial GABA differentially transduce for olfactory adaptation and neuronal aging

Additional comments

1. Figure 3 lacks a figure legend, which is essential for conveying its significance. Including a detailed legend will improve the reader's comprehension of the figure.

2. The manuscript contains numerous spelling and grammatical errors, including those found in the legend for Figure 4. A thorough proofreading is recommended to ensure clarity and professionalism in the final revision.

---

## Round 0.2 · accepted · Accept

Both reviewers are satisfied by the revision. Since all concerns were addressed, the revised manuscript is acceptable in its present form.

Reviewer 1 ·

Basic reporting

The manuscript entitled “Insights and progress on the biosynthesis, metabolism, and physiological functions of GammaAminobutyric Acid (GABA)” in which the authors gave an overview of GABA, focusing on its synthesis, metabolism, GABA receptors, and physiological functions. Also, they discussed the development of ligands for GABA receptor binding site, the prospects of GABA production and application.
The revised manuscript is improved compared to prior revision. My comments were answered and explained by the authors. Therefore, I consider that the revised manuscript is acceptable and suitable for publication in Journal of PeerJ.

Experimental design

The manuscript entitled “Insights and progress on the biosynthesis, metabolism, and physiological functions of GammaAminobutyric Acid (GABA)” in which the authors gave an overview of GABA, focusing on its synthesis, metabolism, GABA receptors, and physiological functions. Also, they discussed the development of ligands for GABA receptor binding site, the prospects of GABA production and application.
The revised manuscript is improved compared to prior revision. My comments were answered and explained by the authors. Therefore, I consider that the revised manuscript is acceptable and suitable for publication in Journal of PeerJ.

Validity of the findings

The manuscript entitled “Insights and progress on the biosynthesis, metabolism, and physiological functions of GammaAminobutyric Acid (GABA)” in which the authors gave an overview of GABA, focusing on its synthesis, metabolism, GABA receptors, and physiological functions. Also, they discussed the development of ligands for GABA receptor binding site, the prospects of GABA production and application.
The revised manuscript is improved compared to prior revision. My comments were answered and explained by the authors. Therefore, I consider that the revised manuscript is acceptable and suitable for publication in Journal of PeerJ.

Additional comments

The manuscript entitled “Insights and progress on the biosynthesis, metabolism, and physiological functions of GammaAminobutyric Acid (GABA)” in which the authors gave an overview of GABA, focusing on its synthesis, metabolism, GABA receptors, and physiological functions. Also, they discussed the development of ligands for GABA receptor binding site, the prospects of GABA production and application.
The revised manuscript is improved compared to prior revision. My comments were answered and explained by the authors. Therefore, I consider that the revised manuscript is acceptable and suitable for publication in Journal of PeerJ.

Reviewer 2 ·

Basic reporting

Overall, the authors have made commendable efforts to incorporate the feedback, resulting in a clear, well-structured, and scientifically rigorous manuscript. I recommend the paper for acceptance in its current form.

Experimental design

iscussion on GABA and Cancer: The inclusion of a detailed section on the role of GABA in cancer, as suggested, has significantly strengthened the manuscript. The authors provided relevant examples and cited recent studies to underscore the therapeutic potential of targeting GABAergic signaling, which enhances the manuscript’s relevance and depth.

Timeline of GABA Development: The addition of a timeline outlining GABA's discovery and key milestones provides historical context and aids the reader’s understanding of its scientific significance.

Incorporation of Suggested References: The authors have included and discussed the references suggested during the review process, further enriching the manuscript with valuable background information.

Validity of the findings

Improved Figures: The authors addressed concerns about figure legends, ensuring clarity and coherence in the visual representation of data. The revisions to Figures 2 and 3 are appropriate and enhance the manuscript's presentation.

Additional comments

Language and Grammar: The manuscript has undergone thorough proofreading, and the grammar, spelling, and overall language have been improved. The editing by a native English speaker has elevated the readability and professionalism of the text.